# Dynamic value alignment through preference aggregation of multiple objectives

## Abstract

The development of ethical AI systems is currently geared toward setting objective functions that align with human objectives. However, finding such functions remains a research challenge, while in RL, setting rewards by hand is a fairly standard approach. We present a methodology for dynamic value alignment, where the values that are to be aligned with are dynamically changing, using a multiple-objective approach. We apply this approach to extend Deep $Q$-Learning to accommodate multiple objectives and evaluate this method on a simplified two-leg intersection controlled by a switching agent. Our approach dynamically accommodates the preferences of drivers on the system and achieves better overall performance across three metrics (speeds, stops, and waits) while integrating objectives that have competing or conflicting actions.

## 1 Introduction

As artificial intelligence (AI) research reaches new peaks, more and more AI systems are being implemented, applied, and deployed worldwide. Further integration of such systems with human societies demands a thorough consideration of their consequences and effects. The inherent property of most, if not all, AI systems is to act with an unprecedented level of autonomy, often in settings where its actions might directly affect human beings. The growing field of Value Alignment (VA) aims to explicitly study the values pursued and exhibited by AI agents and make sure that they correspond to human values.

Motivating examples of VA often consider the long-term and potentially existential threats posed by powerful, super-intelligent AI agents with misaligned values (Russell, 2022a). Not less pertinent are the short-term threats of more mundane, highly specialized AI systems, employed in particular in control settings, becoming misaligned. A prominent case where a potential misalignment is particularly dangerous is given by systems where humans voluntarily cede control of a system to algorithms. Examples of such systems abound: self-driving cars, where the driver cedes control of their vehicle (Haboucha et al., 2017); recommender systems and content algorithms (Carissimo et al., 2023), where the user cedes some control over their access to information; traffic control systems, where drivers cede control of traffic flow coordination (Korecki & Helbing, 2022), are all examples of systems where AI is a control method of choice or is in the process of becoming one[1].

A paradoxical question that arises here is *How do humans stay in control of systems which they voluntarily cede control of to AI?* The paradox can be partially addressed by delimiting the boundaries of what is being controlled and the way in which it is controlled. According to VA, this way should conform to human values, but here we encounter another problem: namely, the challenge of identifying and defining human values. In many cases, it is clear that the values held by humans are varied to the extent of even being contradictory (Awad et al., 2018). Moreover, these values can be situation and time dependent. For instance, road users can value safety, sustainability, travel times, waiting times, and so on, depending on the current state of the traffic and their own conditioning. Thus, AI systems that are truly able to align their values should also be able to do so dynamically, responding to a variety of potentially shifting preference landscapes. This requires a coupling between the AI and the users of the system — the AI must be made aware of the current

---

[1] A contrasting example of such a system, which has not yet been "taken over" by AI is the representative government, to which citizens cede control over many areas of their social lives.

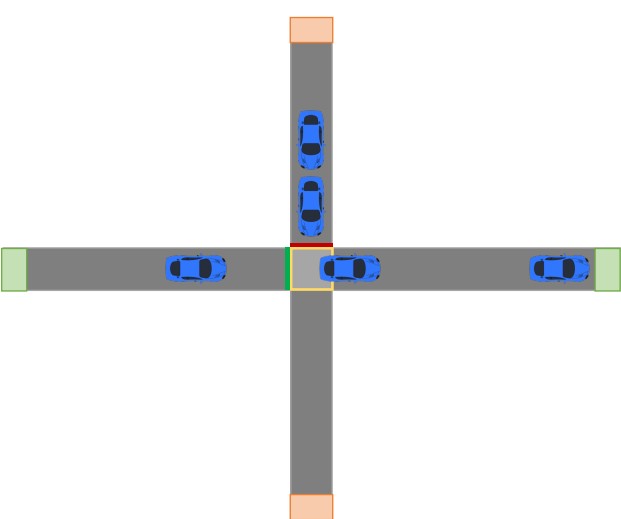

Figure 1: Two-way intersection controlled by an RL agent. The agent controls which direction has a green signal at any given time.

values pursued by the users and be able to incorporate them into its decisions. Among areas that have produced methods and results conducive to such a design are social choice theory, which allows one to gauge and aggregate group preferences, and multi-objective optimization, allowing for a simultaneous pursuit of multiple values. Below, we present a motivational example for the direction of our work and explicitly state our goals.

**Motivating Example**

Consider an intersection with two intersecting approaches as shown in Figure 1. A simple Reinforcement Learning (RL) agent controls the access to the intersection. The agent can switch between two actions — giving green to the North–South approach and red to the West–East approach or, conversely, green to the West–East and red to the North–South. Assume now that the system's designers have chosen to make it control traffic in such a way that it becomes as environmentally sustainable as possible. This can be achieved by setting the reward of the agent to e.g. negative of some measure of emissions or a proxy of it, such as the negative of the number of stops (vehicles emit most emissions when accelerating from complete stops (Rakha & Ding, 2003)). The agent is then trained with this reward and reaches a solution. Namely, it only ever chooses one action, always keeping the red light for the North–South approach and green for the West–East. Indeed, the system has successfully found a global optimum — the emissions, or the proxy of the number of stops, have been minimized. The vehicles on the West–East approach never stop, while the vehicles on the North–South approach stop once, and so the emissions are kept as low as possible (we assume it is impossible to keep all vehicles moving, which is the case if their relative number on each approach is high enough). We are then left with the effects of what could be referred to as "reward hacking" (Skalse et al., 2022) by the RL agent. While the reward selected by the designers has been optimized, what emerges is a probably unwanted and perfectly non-egalitarian solution[2]. This sort of result has been commonly reported in the context of many social systems, where the maximization of social welfare leads to inequality (e.g. in Roughgarden (2002)).

How would that situation develop if the RL agent could incorporate the preferences of users in the system dynamically, rather than always follow a preset and, in this case, clearly misguided goal? Consider a starting population of drivers, all preferring to optimize for sustainability. The system reaches the same state, but now the drivers on the North–South approach are likely to change their preferences — they might no longer value sustainability as highly and switch to preferring to minimize their waiting time. If the RL agent is able

---

[2]The result mentioned here has been replicated in our simulated intersection environment by a Deep Q-Learning RL agent using the negative of the number of stops as its reward.

to incorporate that shift in its decision process, the unfair situation might be escaped or avoided. In this scenario, a certain notion of multi-objectivity can be an asset in avoiding "reward hacking" and the resultant misalignment.

The aim of our work is to investigate a multi-objective and dynamic value alignment in systems where humans cede some control over the system to an AI. We present a simulated intersection environment following the ideas from our motivating example, which can be used to further study the problem. We use a simple traffic environment to test our method and provide a proof-of-concept. A deployment to fully realistic traffic conditions with multiple intersections and complex dynamics is the natural extension of this work. By designing a simplified, but realistic simulation of a real-world system, we hope to provide a more concrete example of the challenges of VA and push this field of study beyond simple, grid-world derived studies, and provide a springboard for extending such systems to more complex problems in the real world. Lastly, we propose a Deep RL agent that can dynamically adapt to different values held by the user of the system under its control. We show how a multi-objective perspective can be leveraged to avoid reward hacking and, at the same time, give more control to the users of the system rather than its designers. We believe that our approach, which explicitly states the social context in which the proposed system exists and takes into account the potential negative consequences of AI systems, is highly relevant with respect to the perceived lack of such considerations in most recent publications (Birhane et al., 2022). Our work builds directly on Multiple-Principal Assistant Games (Fickinger et al., 2020b), which consider the theoretical case of an agent acting on behalf of $N$ humans with differing payoffs. We ground the problem to a particular use case of traffic signal control and extend the methodology to Deep Reinforcement Learning. In terms of arriving at a reward function for the agent to pursue we build on Critch & Russell (2017) and provide a practical implementation that allows for multiple objective to be pursued in real time.

## 2   Related Work

**Value Alignment:** The field has become formalized with respect to AI systems in recent years but followed an established line of research into moral, social, and technical consequences of automation and technology (Wiener, 1960). Presently, the field's motivation often focuses on the risks posed by a hypothetical, powerful, misaligned Artificial General Intelligence (AGI) (Russell, 2022b;a; Hadfield-Menell et al., 2017a). Since there has also been significant criticism of this "singularity" argument (Walsh, 2017) we believe that it is valuable to also consider the problems of VA using more mundane, already existing AI systems (e.g. recommender systems (Stray et al., 2020)). Many researchers support this position, highlighting the lack of substantial work on real-world examples of systems that might suffer from misalignment (Fickinger et al., 2020b; Hadfield-Menell et al., 2017b).

The study of VA is inherently multidisciplinary, bringing insights from the study of ethics, sociology, economics, and AI (Hadfield-Menell & Hadfield, 2019). The philosophical arguments and discussions are well summarized by Gabriel (2020). The more technical side deals with concrete causes of misalignment, such as reward hacking (Amodei et al., 2016). Useful impossibility and uncertainty theorems have also been worked out based on work done by ethicists (Eckersley, 2019).

**Inverse Reinforcement Learning (IRL):** One of the primary methodologies employed for VA is IRL, where the RL agent learns the reward function directly from demonstrations rather than being programmed with it explicitly (Ng et al., 2000). The motivation behind this methodology is learning in situations where explicitly defining the reward function might be challenging but there might exist experts who are able to demonstrate optimal behavior (Abbeel & Ng, 2004). One particular application case has been learning navigation and driving behavior (Abbeel & Ng, 2004; Ziebart et al., 2008). A variety of approaches to the problem of reward learning have been proposed, such as a Bayesian one, where the prior information and the evidence given by expert's actions are used to derived a distribution over the possible reward functions (Ramachandran & Amir, 2007).

Assistance games have been used as a model for the study of alignment problems in much of the IRL literature (Fickinger et al., 2020b; Hadfield-Menell et al., 2016; 2017a;b; Gleave et al., 2022). The setting often involves a robot learning from a single human demonstrator. Fickinger et al. (2020b) extend this setting to multiple demonstrators, and a social choice method is employed to combine the different preferences of

the demonstrators. Brown et al. (2021) proposed a method allowing one to verify and quantify the alignment of a given agent. The issue of goal mis-generalization, where the agent retains its capabilities while pursuing the wrong goal (similar to the motivational example of misspecification given in section 1) has been studied by Langosco et al. (2022) in the context of Deep RL.

The main limitation of IRL with respect to the goal of our work is that it presupposes we can demonstrate the right behavior. This is not possible in many cases, where the agent learns behavior that is not the same as the behavior of the users: e.g. traffic light (agent) learning to control the flow of cars. A further complication is that the values that need to be followed by AI systems are temporal in nature, which is a challenge for this particular methodology. Arnold & Kasenberg (2017) give examples of instances where IRL fails to infer the intended normative behavior.

**Preference-Based Reinforcement Learning (PbRL):** Another methodology, that addresses the difficulty of selecting and engineering the reward function for RL, leverages the preferences (between states, actions, or trajectories) of users or experts (Wirth et al., 2017). What is being learned is a policy consistent with the revealed preferences. PbRL can be deployed in simulator-free (Akrour et al., 2011) as well as model-free settings (Wirth et al., 2016). Some finite time guarantees are possible for tabular settings under the assumption of stochastic preferences (Xu et al., 2020). A set of benchmarks for studying PbRL problems have recently been proposed by Lee et al. (2021). Chen et al. (2022) have been able to extend the PbRL to non-tabular settings for the first time.

While this avenue of research is relevant and informative it is also limited and at the current time infeasible for application to our problem setting. The main issue in PbRL is that the preference landscape is fixed during training (the model learns a given preference landscape), thus limiting the potential for adaptation to novelty in the preference landscapes, which we consider to be a key interest for properly aligned systems.

**Multi-Objective Reinforcement Learning (MORL):** Our work deals with systems whose users might exhibit a variety of potentially contradictory preferences. Thus, a multi-objective perspective is necessary to design an RL agent to work in such an environment.

To study such agents with potentially competitive goals a framework of Markov games has been proposed (Littman, 1994). Algorithms for arriving at a common Pareto optimal policy usually rely on forms of communication between agents (Mariano & Morales, 2000a;b). Some methods aim to learn the entire Pareto front of policies (Van Moffaert & Nowé, 2014). MORL has also been extended to the Deep Learning case (Mossalam et al., 2016).

The main issue of MORL is defining how to combine diverse objectives based on their relative importance. This can be done *a priori*, *a posteriori* or be learned during training (Hayes et al., 2022). In an *a priori* case, the users' preferences need to be specified and fixed, which does not work for our problem setting. Similarly, if the preferences are learned, the model might not be able to accommodate a significant shift in the preference landscape. Thus, in our work, we follow the *a posteriori* approach, where a set of models is trained and the users are able to select between the different models' actions in deployment.

**Social Choice Theory:** The study of aggregating multiple preferences has been consistently identified as a key element relevant to value alignment. The modern history of this field can be traced back to the work of von Neumann and Morgenstern (Von Neumann & Morgenstern, 1947) who applied game theory and other tools to modeling human decision-making (though the origin of the field goes back to the mathematical theory of elections of Borda (1781) and Condorcet (1785) as well as the utilitarianism of Bentham (1789)). The main theories of the field have been summarized in Fishburn (1970) and key impossibility theorems have been given in Arrow (1950).

Some of the works mentioned in the previous sections have explicitly investigated the VA problem through the lens of social choice impossibility theorems (e.g. Gibbard's theorem) (Fickinger et al., 2020a;b). These works have employed insights from the field of mechanism design to circumvent or lessen the effects of potential users misrepresenting their preferences. Methods such as approval voting have been identified as norms that should guide group decision-making for AI systems (Prasad, 2018). Some implementations of such AI voting-based systems have been proposed for ethical decision-making (Noothigattu et al., 2018) based on a large data set of preferences in a trolley problem setting (Awad et al., 2018).

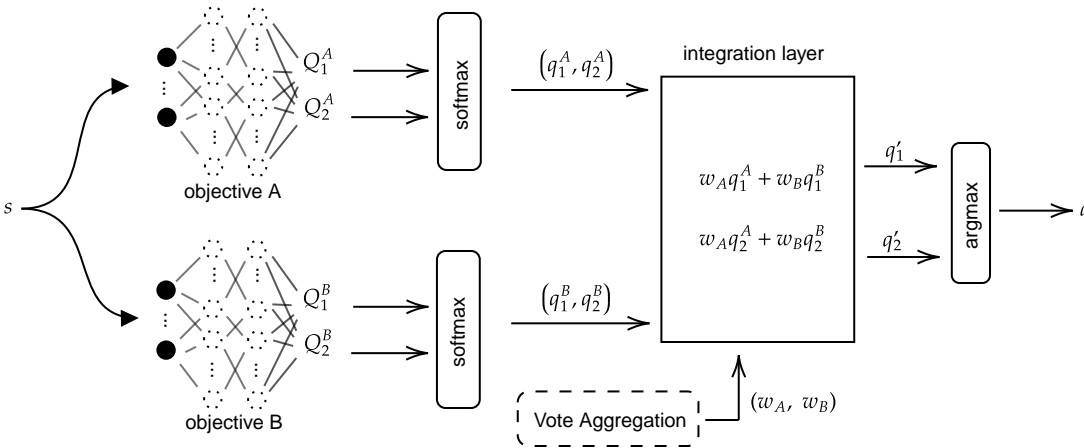

Figure 2: Multi-objective action selection uses multiple Deep $Q$-Networks (DQN) each trained purely on a single objective. For a given environment state $s$, each DQN outputs a vector of $Q$-values. A softmax re-scales these values to reflect their relative importance for a given DQN. These are then passed to an integration layer, where the $q$-values (normalized $Q$-values) corresponding to an action are weighted across the different objectives, using the user votes as weighting coefficients. The resulting new $q'$ values are then used to determine an action, using an **argmax** as in typical DQN methodology.

Our work follows this direction by employing a voting layer directly in the decision-making process of an RL agent. Based on the insights discussed by Baum (2020), our design includes an up-front, explicit way of identifying and aggregating potentially contrary preferences rather than off-loading that process to the RL agent (to let it figure it out so to speak). In the following section, we give details of our approach and test its efficiency under two voting schemes.

## 3  Methodology

Our proposed approach has three main components: the models of different objectives, a method of vote aggregation, and an integration layer. Here, we describe in detail the employed design principles. We focus on control settings where humans cede some control over the systems to the AI agent, which here is assumed to be a Deep RL agent. Finally, we introduce a simulated environment (inspired by our motivational example) in which we test our approach. In the following sections when we refer to the agent we always mean the controller, in our case traffic signal controller. We refer to the users of the systems as users, drivers, or vehicles. The users are not learning, while the controller takes its decision based on models learned through $Q$-learning.

**Multiple Objectives Models**

As we are interested in the issues of dynamic value alignment to multiple values in the context of control, we need a way to model the effects of different actions on the state of the system with respect to different objectives. We propose to train a separate Deep $Q$-Network (DQN) for each of our objectives. The DQN returns a $Q$-value for a given state–action pair, which can be interpreted as the measure of the expected future discounted rewards if the given action is taken in the given state.

Consider a DQN for an objective $A$ with two possible actions. At inference, in some state $S$ the DQN will output $Q_1^A$ and $Q_2^A$, denoting the expected reward. Typically, the action corresponding to the larger of these two $Q$-values will be preferred and selected. However, since these $Q$-values are interpretable, one could also take into account their relative quantities. Namely, if $Q_1^A \gg Q_2^A$ or $Q_1^A \ll Q_2^A$ there is a clear preference for one of the actions. On the other hand, if $Q_1^A \approx Q_2^A$ then it might not make much difference which action is chosen even if one of the $Q$-values is technically greater than the other. We will develop this observation

further when discussing the integration layer. Moreover, we note that the $Q$-values could also be compared between DQNs trained with different objectives if the two DQNs share the same action space. Specifically, $Q_1^A$ could be compared with $Q_1^B$, where $B$ represents another objective. This is limited in that if the scale of the objectives is not the same, a fair comparison becomes more difficult. However, since here we will care only for the relative importance of the $Q$-values of a given objective $(Q_1^A, Q_2^A)$ we can normalize them (in this work we use a softmax normalizing function). These normalized values we refer to as $q_1^A$, $q_2^A$ for objective $A$. Since their scale will be in the range of 0 to 1, we are now able to compare $q_1^A$ with $q_1^B$.

**Vote Aggregation**

We want to allow the users of our system to be able to reveal their preferences and affect the system according to them. Therefore, we need to specify the type of preferences such that they are meaningful to the system. Here, we set the possible preferences to correspond to the reward $r$ of the DQNs described in the previous section. Moreover, we say that when a user selects a given reward as the preferred one, they do so under the assumption that this reward will be applied to the entire system and not preferentially to them alone. If the preferred $r$ corresponds to minimizing the waiting time, the user asks the controller of the system that everybody's waiting time be minimized and not just their own[3]. Thus, the users remain under the *veil of ignorance* (Harsanyi, 1953) as the exact effects of their chosen objective being pursued by the controller are not known to them. Given this setup, the users of the system are polled at decision time (each time the traffic signal control agent is to take action, in our environment that happens in 5-second intervals) and reveal their preferences. At the decision time, only users that are allowed to vote at the given moment are polled—these are the users on the incoming lanes of the intersection. Thus, the voting population at each step is not equal to the entire population of the system. Even though the preferences of each user are fixed, the relative preferences of the voting population change as the population changes. These preferences need to then be aggregated and weights $w$ for each of the possible preferences are created. In this work, we consider and compare two simple aggregation methods.

**Majority voting:** The option with more votes ($v_A$ being the number of votes for preference $A$) wins, and only the winning option is considered.

$$w_A, w_B = \begin{cases} (1,0), & v_A > v_B \\ (0,1), & v_A < v_B \end{cases} \tag{1}$$

Thus, if preference A gets more votes, $w_A = 1$ and $w_B = 0$, for a case with two possible preferences A and B.

**Proportional voting:** The weights are created directly from the votes according to Equation 2.

$$\begin{aligned} w_A &= \frac{v_A}{v_A + v_B} \\ w_B &= \frac{v_B}{v_A + v_B} \end{aligned} \tag{2}$$

**Integration Layer**

The key element of our approach is the integration layer, which allows for the integration of the DQNs results with the preferences of the users. The layer affects the integration according to Equation 3.

$$\begin{aligned} q'_1 &= w_A q_1^A + w_B q_1^B \\ q'_2 &= w_A q_2^A + w_B q_2^B \end{aligned} \tag{3}$$

The $q'_1$ and $q'_2$ values encapsulate the expected rewards weighted by the weights $w$ for the two possible actions. In practice, in cases where there is a strong preference (e.g. $w_A \gg w_B$) the action chosen under

---

[3]"Act only according to that maxim whereby you can at the same time will that it should become a universal law." –Kant (1785)

objective $A$ (say $q_2^A > q_1^A$) will be selected unless $q_1^B \gg q_2^B$ and $q_1^A \approx q_2^A$. Thus, the system generally follows the preferences unless the relative difference between the expected rewards within one objective is large and within the other is small. Conversely, if the preference is weak ($w_A \approx w_B$) the system will prioritize the objective that stands to lose more from not being followed. This approach then allows accounting for user preferences, but at the same time tends to avoid taking very bad actions on one of the objectives.

Here we have described a version of our method with two objectives, but our system allows for more objectives to be included. One could imagine new objectives to be continuously added, as long as it is possible to train new DQNs based on different objectives.

The approach we have described throughout this section is summarized symbolically in Figure 2.

**Switching Traffic Lights**

We modeled a simplified two-leg intersection that consists of two one-way roads intersecting at one point (see Figure 1). The intersection has traffic signals controlled by a switching agent. The switching agent facilitates the assignment of a green light to one of the roads and a red light to the other. Regularly, at intervals of $t_{\mathrm{act}}$, the switching agent may choose to switch the active phase on the intersection. When choosing to switch or not, the agent polls the vehicles on the upstream approach (in other words only vehicles that will be directly affected by this action are allowed to vote, the vehicles on the downstream do not vote).

We train two Deep $Q$-Learning models on two different reward functions: stops and wait times. The state of the traffic signal control agent (and so the state of the model Deep $Q$-Network) is the occupancy of the vehicles (which is a function of the number and size of the vehicles) on the three segments of the incoming approaches (an approach is divided into three segments of equal length).

$$r_{\mathrm{stops}} = - \text{ (number of stops in the past } t_{\mathrm{act}} \text{ steps)} \tag{4}$$

$$r_{\mathrm{wait}} = - \text{ (duration of stops in the past } t_{\mathrm{act}} \text{ steps)} \tag{5}$$

We also considered the case of using a single reward function to incorporate multiple objectives simultaneously. We train two additional Deep $Q$-Learning models on two different reward functions: a linear combination of stops and wait times (Equation 6) and Cobb–Douglas production function of stops and wait times (Equation 7).

$$r_{\mathrm{linear}} = \alpha \, \mathrm{norm}(r_{\mathrm{stops}}) + \beta \, \mathrm{norm}(r_{\mathrm{wait}}) \tag{6}$$

$$r_{\mathrm{cobb\text{-}doug}} = - \, \mathrm{norm}(-r_{\mathrm{stops}})^{\alpha} * \mathrm{norm}(-r_{\mathrm{wait}})^{\beta}, \tag{7}$$

where norm is the normalizing factor that scales the given value by its maximum for the stops and by half of the maximum for the wait times, and $\alpha = \beta = 0.5$.

**Experiments:** To test the performance of our approach, we run experiments in the simulated environment described above. We run the system under six different demand distributions. The demands are defined in terms of the number of cars on each of the two approaches (N, W), where N represents the number of cars on the North–South link and W represents the number of cars on the West–East approach. The demand can be: low unbalanced (11, 6); low balanced (11, 11); medium unbalanced (22, 11); medium balanced (22, 22); high unbalanced (32, 16); high balanced (32, 32).

For each demand, we set the preference of the participating vehicles to be half-half — 50% of them prefer $r_{\mathrm{stops}}$ and 50% prefer $r_{\mathrm{wait}}$. Depending on the random initialization and the system dynamics, the proportions of the voting user's preferences change throughout the simulation. We ran our method 100 times for each demand setting with different random initialization and presented averaged results.

We conducted all experiments using the CityFlow simulator (Zhang et al., 2019) and we used periodic boundaries for the two roads. Each scenario is run for 3600 steps, corresponding to an hour of real-time. The code used for all the experiments and results (including all the hyperparameter settings) is available at this repository.

**Underlying Assumptions**

The expected performance of the proposed method rests on some assumptions, which we make explicit and discuss in this section. The first assumption is that of a centralized controller. We have stated in section 1 that in this work, we are interested in systems where users voluntarily cede control. As such this control needs to be ceded to some particular decision mechanism, which is assumed to be centralized. This stems from our particular application to traffic signal control. The traffic system users are assumed to follow the common (though perhaps implicit) agreement on traffic laws relying on the existence of traffic lights, which are inherently a form of centralized control. From this assumption, another one follows: namely, that the preference aggregation rule used by our method is socially trusted. This assumption is akin to the current trust (or lack thereof) of road users in traffic signal control algorithms. Further assumption relating to the preference aggregation method is that it is well evaluated from the utilitarian perspective. While our method allows for any aggregation method to be employed, the analysis of the effects and trust in different ones is out of the scope of this work (thus, we rely on the above assumptions). It is worth noting that the practical side of implementing such a system would likely need to include consultation and educational efforts to inform the users about how it works.

A further assumption that we take is that of selfish rationality and truthfulness of traffic users. We assume that the voting users do indeed vote according to their real preferences. A common problem that could affect the system is strategic voting, whereby users misrepresent their preferences to exploit the aggregation method and increase the chances of their objective being chosen (Gibbard, 1973). This problem occurs mainly in schemes such as cumulative voting, where the user expresses their preferences as a ranking. In this work the consequences of the theorem are avoided since it only applies to cases with three or more options (we only have two options in this work). Therefore, in this work the strategic behavior of users would be equivalent to honest behavior. It is however worth noting that if this approach was applied to a situation with three or more objectives one would have to take into account the problem of users potentially mis-representing their preferences.

## 4 Results

### 4.1 Reward functions with single objectives

In Figure 3 we report the performance on three different metrics of the two single-objective DQNs (optimizing for Equation 5) and the two multi-objective methods, one using majority voting and the other proportional voting. We observe that the speeds do not vary much between the methods (while they do vary between demands). It is also apparent that the single-objective DQNs suffer a trade-off between favorable stops or wait times. The Stops DQN is able to achieve a very low number of stops (close to 0 for all scenarios) but at the same time, its waiting times are prohibitively large. In fact, the Stops DQN learns to never switch, thus reaching a global minimum of the number of stops in systems where free flow is impossible, but making one lane of traffic wait forever. Conversely, the Wait Times DQN achieves very low wait times (close to 0 for all demands) but a high number of stops (qualitatively this solution corresponds to the agent switching all the time, alternating the green signals between two lanes cyclically). The multi-objective methods, on the other hand, never achieve performance as high as the single model on each objective. However, they are able to achieve good results on each of them — effectively, they are able to pursue and find a balance between them both. The Proportional method outperforms the Majority method in all demands except Medium Balanced (where they are the same) and High Balanced (where Proportional achieves fewer stops and higher wait times than Majority).

### 4.2 Reward functions combining multiple objectives

In Figure 4 we report the performance on three different metrics of the two single-objective DQNs aggregating two objectives into one and the proportional voting-based multi-objective method. We note that in the Low conditions the method performs worse than the Cobb–Doug, which achieves both better wait times and stops. In the Medium Unbalanced condition the Prop S+W performs on the same level as Cobb–Doug in all metrics. In the Medium Balanced the Prop S+W performs in between the two methods in terms of stops

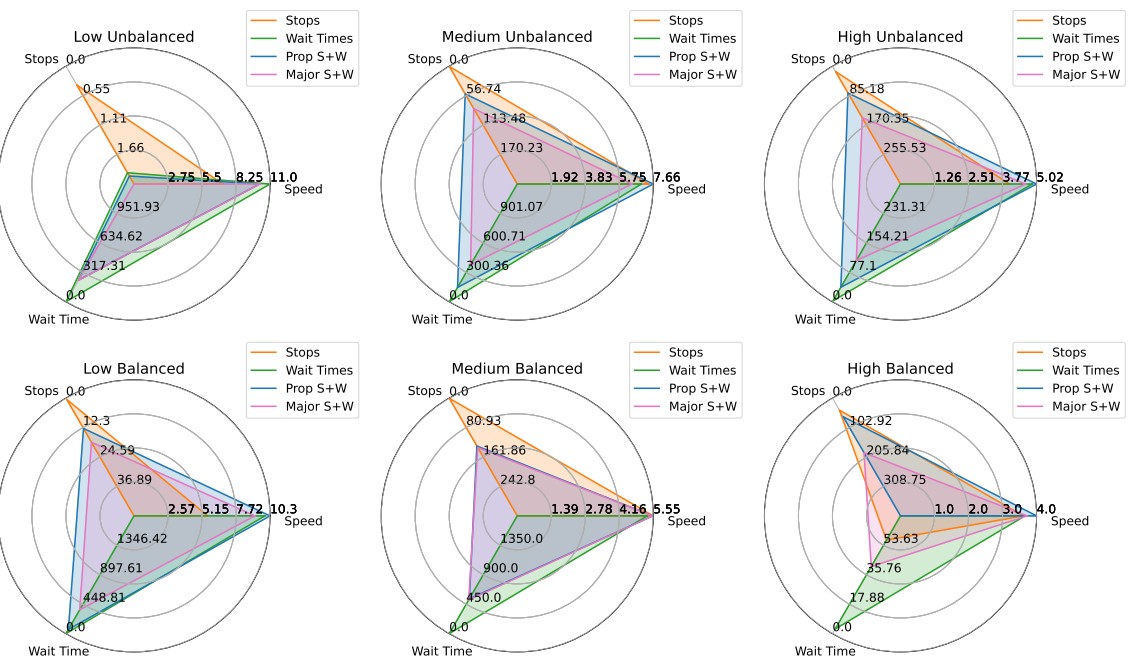

Figure 3: Radar plots for the three evaluation metrics (stops (seconds), wait times (seconds), and speeds (meters per second)) for six different demand distributions. Larger triangle areas can be associated with a more desirable performance across all metrics. The scales are different for each radar plot. Prop S+W represents our method using proportional aggregation, Major S+W represents our method using majority aggregation.

and wait times. High Unbalanced and High Balanced conditions, arguably the most challenging ones, are where the Prop S+W shines, achieving performance at the level of Linear comb. and Cobb–Doug in both metrics at the same time while each of the two methods underperforms on one of the metrics.

### 4.3 Alignment of the multi-objective method

In Figure 5 we report the correlations between the actions chosen by different DQNs in a given state. We observe that a DQN optimizing for $r_{\text{stops}}$ has a low correlation with a DQN optimizing for $r_{\text{wait}}$. This means that these DQNs tend to predict different actions as optimal given the same state. We see however that the multi-objective actions can align with either $Q^{\text{stop}}$ or $Q^{\text{wait}}$, for different demand conditions.

## 5 Discussion

Complex systems often have numerous objectives that designers may fail to account for during the design stage. Our proposed method offers modularity in accounting for novel preferences, making it well-suited to accommodate new objectives even after the system is deployed. By training a Deep $Q$-Network or any other model that outputs $Q$-value estimates, we can optimize the system for new objectives as needed. That is, once the system is deployed, users, designers, or controllers of the system can easily add new objectives once they realize the need for adjustment. Thus, one could imagine a system where users can indicate the objectives/preferences they care about that are currently unavailable. Models optimizing for these objectives could then be trained in simulation or using historical data and deployed to the system. Users would then be able to vote for these preferences, and they would be taken into consideration proportionally to the vote. The central aspect allowing for this is that the system is able to work with any number of objectives. Nevertheless, we believe a further study (which is beyond the scope of this paper, whose goal is to introduce this kind

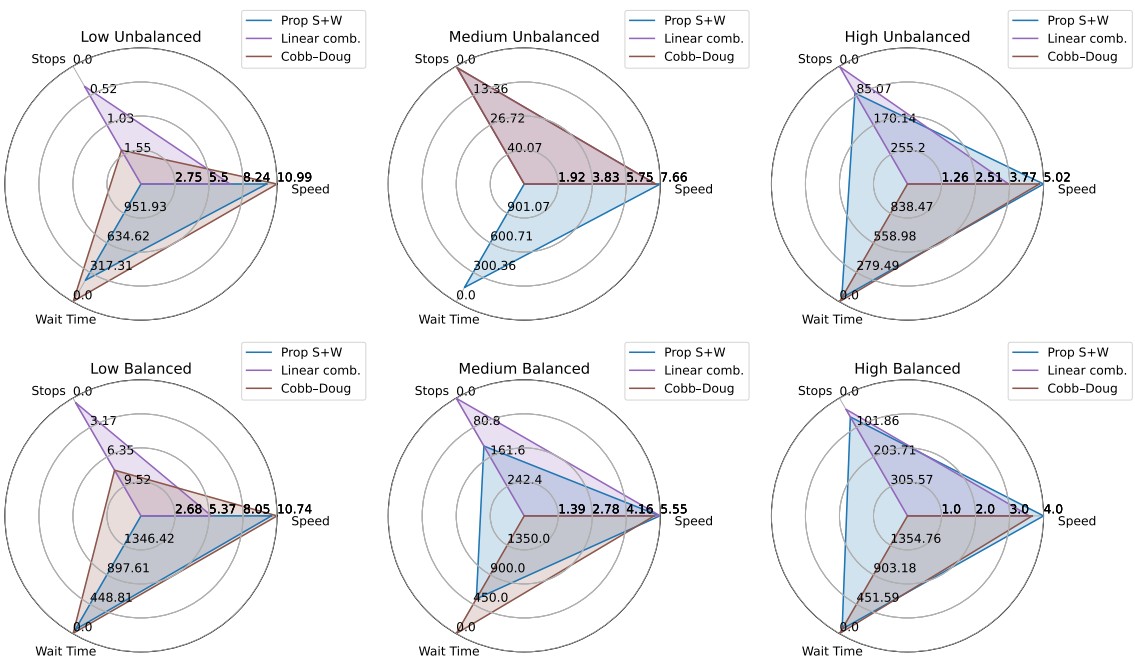

Figure 4: Radar plots for the three evaluation metrics (stops (seconds), wait times (seconds), and speeds (meters per second)) for six different demand distributions. Larger triangle areas can be associated with a more desirable performance across all metrics. Here, we compare the multi-objective approach (Prop S+W) against two single-objective approaches that incorporate two metrics (Linear combination and a Cobb–Douglas product function). The scales are different for each radar plot.

of voting-induced multi-objective methodology) into the feasibility of including more than two objectives is needed.

In this traffic-inspired simulation environment, one could argue that designers can choose reasonable objectives so as to avoid the suboptimal/unfair policies learned through an inappropriate reward function. However, in a broader sense, there could also be systems with many incommensurate rewards, and setting them *a priori* can amount to a sort of algorithmic dictatorship. Moreover, in many cases it is impossible currently to account for all the potential ways in which the reward can become hacked or misaligned and so setting fixed objectives carries a significant risk of running into misalignment at some point.

Using multiple single-objective reward functions (our approach) comes with some tradeoffs compared to using a single multiple-objective reward function. Our approach requires implementing separate $Q$-value estimate models for each objective. While the number of potential objectives varies across different application domains, we believe that in most cases, the number of objectives would not be too large, and implementing a few Deep $Q$-Networks (DQNs) should not be a significant limitation. However, for problems where a large number of objectives is desired, scaling issues may arise when using more traditional Pareto-front based methods. Multi-objective optimization may have difficulty scaling to a large number of objectives. Alternatively, attempting to optimize a linear or non-linear combination of multiple objectives using a single model (e.g., a deep neural network) would require specifying the parameters such as the weights given to each objective. We note that in our experiments optimizing the linear combination of the two objectives actually leads the system into the undesired optimum of the close to zero number of stops but very high wait times. This showcases the potential effects of misspecifing the weights. Moreover, if an additional objective is added to the model, it would require retraining the model from scratch, and once again searching and specifying the weights for the objectives. Our approach avoids these challenges, although it requires training additional single-objective models. Overall, we believe that our approach is a more scalable alternative to the method outlined above.

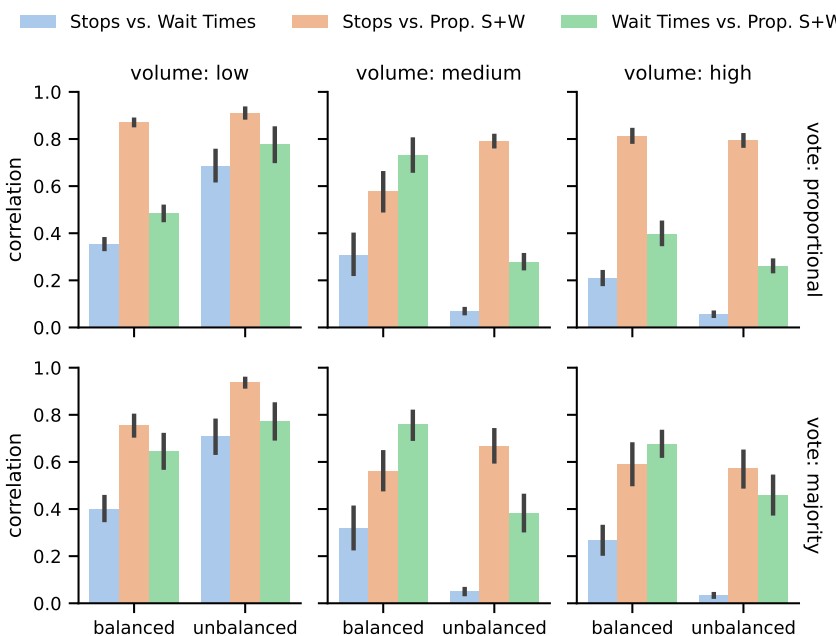

Figure 5: Correlation of the actions that would have been chosen by the Stops ($r_{\mathrm{stops}}$), Wait Times ($r_{\mathrm{wait}}$), and Multi-Objective (our method) DQNs in a given state for the different traffic volumes (low, medium, high) and voting schemes (proportional and majority vote). Higher values mean the same action would have been taken by both schemes.

### 5.1 Limitations

Our approach has a unique requirement for an environment: the environment must have multiple players (with their own logic) that interact with a system that is controlled and optimized by a controller. To our knowledge, additional environments would need to be made from scratch, which is why we have focused on only one environment.

Another limitation of our work is that we focus only on a half-half split between the two possible preferences. While due to the dynamics of the studied system and random initialization, we still get a variety of proportions in the population voting in each action phase, it would be interesting to further study the performance of our approach under different preference splits.

Furthermore, the approach we propose relies on optimization. As such it might suffer from limits of optimization such as potential incomparability of alternatives (Carissimo & Korecki, 2023). In practice it would make sense for the objectives that are available to be voted on were themselves selected in a democratic and participative way. A limitation of our work is that we assume these objectives to be given and at least in some sense commensurable (to the extent that they can be voted on).

Moreover, we test our system with two objectives; a clear extension would be to run experiments with three or more objectives. However, we show that combining even two objectives can have a positive effect and help avoid the negative effects of missepcified rewards. Lastly, a more formal quantification of the alignment of our approach would be valuable.

## 6    Conclusions

Based on our results, we can confirm that our proposed method avoids the "reward hacked" solution exhibited by the Stops DQN. Thus, our method is able to avoid misalignment in our setting, by following a more nuanced, multi-objective perspective. Moreover, our method performs well on both objectives, aligning well

with both preferences exhibited by the user population. Furthermore, the social effects of deploying such a participatory system cannot be quantified in a simulation but are likely to be positive, promoting engagement and trust, and giving more control over the system to its users. We also note that the proportional voting system appears to work better in our setting than majority voting. This appears intuitive as the proportional method mitigates issues such as "tyranny of the majority" and "wasted votes", which have been associated with majority voting methods. An interesting direction for future work is modeling the users of the system (vehicles) as learning agents, who can change their preferences depending on their experiences.

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
