# OpenReview forum: "Dynamic value alignment through preference aggregation of multiple objectives"
_TMLR — Rejected by TMLR_

### Review · Reviewer_Fnpy · 2023-05-22

**Summary Of Contributions:**

The submission discusses the challenges of dynamically aligning the value of a machine system to a changing population whose constituents posses changing and incompatible preferences. It proposes a method of dynamically aggregating constituent preferences and tests this approach on an intersection problem.

**Audience:**

Yes

**Claims And Evidence:**

Yes

**Requested Changes:**

I think the work is sufficient for acceptance as is.

**Strengths And Weaknesses:**

Overall, I liked the submission. Here are some constructive criticisms I had while reading:
- The introduction feels a bit unnecessarily flowery.
- I thought the background information on the problem was nicely written. One thing I wondered about which I didn't notice discussed is whether there ought to be game-theoretic considerations in these settings (i.e., whether a user is incentivized to disclose their true preferences / associated consequences).
- I personally find the radar plots a bit hard to read.
- One thing I wondered about is how the intersection policy changed over time for the adaptive methods (i.e., whether it oscillated/cycled or not).

---

> ### Author Response · Authors · 2023-06-22
> **Response**
>
> Thank you very much for your positive review, we were very happy to read it. We have made some edits to the introduction, the background, and a new section in methods (underlying assumptions) which we believe address some of the issues raised by the reviewer.

---

### Review · Reviewer_vGax · 2023-06-08

**Summary Of Contributions:**

The article presents a system for automatically choosing actions on behalf of agents interacting in an environment where there might be multiple, potentially incompatible, preferences over outcomes across the agents. The authors use multiple Q-learning networks and combine their recommendations together according to some voting system to provide the final actions that the controller will implement. The work is motivated by a traffic intersection environment where the controller is in charge of the traffic light, and the agents vote to either minimise stops, or to minimise waiting time.

**Audience:**

No

**Claims And Evidence:**

No

**Requested Changes:**

The article would benefit from a few categories of broad change:

1. Provide appropriate references placing the work in the context of the relevant fields. This includes discussion of prior work on Game Theory, Philosophy and AI
2. Provide enough details about who are the agents, who is learning, what are their inputs, outputs and incentives
3. Make a more realistic example, or drop the unnecessary RL framing

## More specific comments

### In Motivating Example

The authors claim that a controller minimising emissions would only make one side of the intersection go, and make the other wait forever. They argue this is true because one can use the number of stops as a proxy for emissions. But this is obviously a gross oversimplification, and highlights the overall tone of the paper. There is extensive literature on poorly or wrongly specified goals, and this is a real issue when designing artificial systems. However, the authors then go to claim that their system resolved this issue (presumably because their system alternates the green/red lights), even though, as discussed above, it was quite heavily engineered to provide alternation.

In addition, the authors claim that the optimised solution of never changing the light is unwanted and perfectly non-egalitarian solution. This is an observation from a wide variety of fields, where maximal social welfare is at times only achievable through inequality. For instance, the wide debate on Rawlsian versus Kantian morality, and the implication to ML systems. Please contextualise these claims appropriately.

### In Preference-based RL

The authors claim PbRL is not applicable to their setting, but one could easily imagine providing preferences as inequalities, or constraints which would enable direct comparison with the methods proposed.

### In MORL

The authors claim that a learned preference would not be able to accommodate significant shift. This would require some strong justification. For instance, Hierarchical-RL is specifically tailored to having disparate goals represented and deployed flexibly into a hierarchical policy, and is a learned system.

### In Multiple Objectives Models

The authors train a different Q-Network for each of the goals. This would not generalise to situations where history matters, since following one policy induced by one Q-Net would end up in state spaces where another Q-Net is too out of distribution. A better approach would be to train a system that has experienced shifting goals at training time. For instance, again, something like HRL. This doesn't matter for their example environment due to its simplicity, though.

### In Vote Aggregation

The authors refer to reward functions simply as rewards, and this can be confusing. Perhaps is better to say goals, instead to avoid confusion?

The authors bake in the assumption that a vote for a goal must necessarily apply to everyone. The claim that this is to prevent selfishness is puzzling (as explained above, even in these systems selfish agents can exploit). Moreover, it's not clear that authors abide by their own rules, since only some agents are allowed to vote, and those votes will directly impact the other agents in an adverse effect. Perhaps this would make more sense if the voting were behind a Veil of Ignorance, or if the authors just don't make a claim of universality.

### In Figure 3

The caption does not explain the meaning of Prop S+W nor Major S+W. The multiple scales and radar plots are an odd choice; they help within relative comparisons, but mask the extreme-ness of the different setups.

### In Limitations

The section only talks about superficial limitations: having to design an environment, a specific arrangement of agents, number of objectives. What I would like to know is: does it scale (in time, in population size, in environment complexity)? What are the limitations of preference functions? How bad is it when voting preferences are not honest?

**Strengths And Weaknesses:**

# Strengths

The article highlights an important aspect of current and future uses of technology where individuals are unable or unwilling to make complete decisions for themselves, and instead use technological proxies to make decisions on their behalf, and where preferences might be heterogenous.

The article is written in an accessible way.

The ideas are explored via an environment of limited complexity, but it is possible that with some adaptation they could be applicable to more complex situations.

# Weaknesses

The work suffers from a range of limitations and render it of limited interest. At the highest level, it is not clear that the framing of the problem as a reinforcement learning problem with learning of policies is appropriate. For instance, there is no state in the system beyond where the agents (cars) are, and at least the average behaviour can be captured by a Markov Decision Process, possibly the whole system. Since goals are binary and decisions of the controller are also binary, this whole setup could be done tabular, depending on what the observation of the controller might be. Individual agents are not learning in any significant sense (other than possibly changing their preference, although not enough information about inputs for the controller nor agents are given). It is never explained how the agents might change their preferences, other than to say they do so stochastically.

Their motivating system is of sufficient simplicity that there is really no need to conduct learning at all, and I highly suspect that closed form solutions could be easily derived providing optimal control.

The aggregation of goals by simple normalisation is mathematically convenient, but loses most of the interesting nuance of actual real-world value alignment or consensus driven decision making. The authors discuss how vote aggregation might impact the dynamics of the system. They claim that their method prevents the controller from "reward hacking" a solution that optimises, say, environmental cost (as proxied by number of stops). Yet, their system requires that only the "affected" individuals vote on changing the controllers goal, and alternates between the one on one side of the intersection to the other. This already bakes in _way too much_ into the decision. Worse still, if the voting aggregation function is known, it could be exploited by malicious agents. For instance, if honest agents only have a slight preference between commute time versus environmental impact, and the controller is using proportional vote aggregation, then a malicious (cohort of) agent(s) might overstate their preference for minimisation of commute time, skewing the controller's decision in their favour. This is a well known phenomenon in Game Theory, and is never discussed. Generally, when reward for different goals have very different scales, combining those goals is not trivial, and could be fundamentally impossible to unidirectionally resolve (e.g. the Trolley Problem). The authors even highlight this issue in the first page of their paper, never to discuss it again in the context of their contribution.

The work is very colloquially presented, to the point where important details are omitted, and critical references are absent. The first page makes a number of bold claims, and has a total of *2* citations on it! At the top of the second page, the authors are highlighting how some fields have considered the problem at hand, but not only do they not provide any references, they also neglect very large swaths of the literature, including: the veil of ignorance, Kantian ethics, consequentialism, etc. on the philosophy side; and Mediated equilibria and other solution concepts in Game Theory.

---

> ### Author Response · Authors · 2023-06-22
> **Response**
>
> We thank the reviewer and would like to engage with the comments made. Let us address the changes that were requested first (changes in the new version are in blue).
> 1. We have already included many references to game theory in the context of value alignment, IRL, and MORL. There is a large number of game theoretic works that we cite (such as The off-switch game or Theory of games and economic behavior and more). As such we do not believe there is a need to have a dedicated section on game theory. We motivate it further by the fact that our work does not focus on a game theoretic analysis of the problem, which has been conducted in the cited work, but rather on the issues of practical implementation in real systems.
> We cite works that report on philosophical issues that are relevant to value alignment, such as Iason Gabriel. Artificial intelligence, values, and alignment. Minds and Machines or much of the work cited in the social choice theory section. Here we also believe that a separate section is not needed as our work's goal is not to delve deeper into the philosophical underpinnings (which is not to say that these are not very relevant and worth-researching topics). Where appropriate we try to employ a certain level of ethical awareness (etc. using veil of ignorance in our design) but ultimately the system we propose relies mainly on the ethics of its users.
> We hope that the reviewer understands our motivation for not creating separate sections (as it is we already have 5 subsections in the background) but rests assured that we have included relevant citations on philosophy and game theory as relevant in value alignment. If we have, however, missed some key work we would be very happy if the reviewer pointed it out to us so we might include it.
> 2. When we refer to the agent we always mean the traffic signal control agent and the drivers in the system are referred to as users, vehicles or drivers. The traffic signal control agent uses a learned model (the agent is not learning in real time but is trained off-line and afterwards deployed). We have also reworked the methods section to include information on the state and design of the agent.
> 3. The reason we include RL in our study is that the value alignment problem itself arises in learning systems and has been studied in relation to RL systems a lot. Thus by also using an RL agent we can refer to and build up on the existing literature in the field. While we agree that our system is simple we propose it as a first step that allows us to validate our methodology and provide a proof of concept. The natural next step that we are already working on is applying our methodology to a larger, more realistic traffic system. A further reason we use a simplified system is that our methodology is quite complex, thus testing it on a simple system gives us a certain ease in drawing conclusions about the control method itself. We agree that the next step is to apply it to a more realistic setting.
>
> Further questions:
> * The drivers are not learning. They also do not change their preferences but the setup of our experiments and voting does make it so that the controller agent faces voting populations with different proportions of preferences.
> * Indeed we agree that it would be possible to derive optimal control with respect to the single objectives --- for the simple system that we consider it is possible in most scenarios to achieve a number of stops equal to 0 or wait times equal to 0 (and the single objective models that we learn do reach these optimal policies). It is however not clear how to derive optimal control with respect to multiple objectives where preferences over them expressed by the voting population change due to the dynamics of the system.
> * We are not sure if we understood what the reviewer means by "Yet, their system requires that only the "affected" individuals vote on changing the controllers goal, and alternates between the one on one side of the intersection to the other."
> What we meant by allowing only affected individuals to vote is that only the vehicles on the incoming lanes of the intersection are allowed to vote (this includes both the N-S and W-E approaches). We do not in any way "bake in" the alternation between the lanes into the system.
> * We added a section that addresses the reviewer's reservation about the voters potentially misrepresenting their preferences (section Underlying Assumptions).
> * We added a paragraph in the limitations where we discuss the problem of combining potentially incommensurable objectives.
> * We extended the introduction with references.
> * We have contextualised the claims we make in the motivating example as suggested.
> We hope to engage in further dialogue with the reviewer. Since we only have limited time to answer the reviews we have tried to address the most pressing issues first. We are happy to discuss and apply further changes if it is deemed productive by the reviewer.

---

### Review · Reviewer_gaay · 2023-06-18

**Summary Of Contributions:**

This paper investigates how to adapt algorithmic agents to reflect the values of participants in the automated system by incorporating a voting module whose outcome is used to reweight the preferences. It demonstrates that the resultant agent provides a better balance of the original single-objectives used to train the agents.

**Audience:**

Yes

**Claims And Evidence:**

Yes

**Requested Changes:**

The biggest change I would suggest, and one that would make me reconsider, is to expand the scope of this paper to make clear what new challenges and unsolved problems are being introduced by moving from gridworlds to another system. The system introduced here is fairly simple and as a consequence seems like it could be analyzed without any simulation so I urge the authors to identify either a system or a problem whose dynamics or resolution is unclear and demands further investigation to understand properly. I was not able to figure that out from reading the paper. What is distinct about the system being studied here that makes it different from studying gridworlds?

This is also just a suggestion and not something that would affect my review but the paper makes references to some not easily quantified benefits of using a participatory scheme in the conclusions section. This seems like an interesting question worth studying! How do humans prefer to have their opinions taken into account when participating in such a system.

I think it would be useful as an expository tool to expand on the voting section to explain how the votes are converted into weights. For example, in the experiments section they say that the population has a 50% preference over stops vs. weights. Does this thing imply that $v_a$ and $v_b$ are both 50 and so $w_a$ and $w_b$ will be 50? I think what I wasn't clear on was whether the agents were voting directly for a particular reward function or whether the voting was done online in response to the state of the simulator. I suspect it's the former and not the latter. The latter would have been interesting though is not necessary to change how I feel about the paper.

Nitpicks:
- "A growing field of Value Alignment (VA) aims to explicitly study" should be "The growing field"

**Strengths And Weaknesses:**

The paper is very well written and tackles an interesting problem. I think it is exciting to see the study of aligning larger scale systems with human preferences

The most glaring weakness to me is that it is not clear what question, either scientific or engineering, is being answered by this paper.  Is the point that if you have two misaligned agents, they can be combined to create a better agent by weighting the two?
Additionally, it is not clear what is added by the inclusion of RL or the study of a time-dependent system. It feels as though the results of this paper would have been equally valid and could be completely understood by studying the equivalent stateless matrix game.

I suspect the paper needs a very significant rewrite to make clearer what its contributions are.

---

> ### Author Response · Authors · 2023-06-22
> **Response**
>
> We appreciate reviewer's comments on our work, the points raised are well-taken. Let us answer here as to what we have changed to satisfy the issues that were pointed out (all changes are in blue in the new manuscript).
>
> We have expanded our introduction and methodology section to further clarify what the purpose of this work is. We summarise the key elements here so as to answer reviewer's questions. The reviewer asks for clarifying the reason for moving from gridworlds to our traffic system. We believe that this in itself is part of our contribution - gridworlds are highly artificial and do not have any inherent social dimension (which is a key consideration in value alignment), our system is inherently social and grounds the value alignment problem in more concrete terms. Many of the papers from value alignment literature point to the need of giving more realistic examples and showing the value alignment problem and its solutions in systems other than gridworld.
>
> While we agree that our system is simple we propose it as a first step that allows us to validate our methodology and provide a proof of concept. The natural next step that we are already working on is applying our methodology to a larger, more realistic traffic system, which certainly cannot be approximated easily using games or other such models. We would also like to ask the reviewer if they could perhaps clarify what stateless matrix game they mean exactly as we do not see how our traffic signal control agent or the traffic system could be replaced by such a game.
>
> Furthermore, the reason we include RL in our study is that the value alignment problem itself arises in learning systems and has been studied in relation to RL systems a lot. Thus by also using an RL agent we can refer to and build up on the existing literature in the field.
>
> We also clarified and fixed some of the nomenclature used in the paper that we think might have not been clear. When we refer to the agent we always mean the traffic signal control agent and the drivers in the system are referred to as users, vehicles, or drivers.
>
> As per the reviewer's request, we have expanded the voting section to explain how the votes conversion happens. The drivers vote on the particular objective at every decision step which is set to be 5 seconds in our simulation. Only the drivers on the incoming lanes are able to vote and so even if the global population has 50-50 preference the voting population might have different preferences depending on the dynamics of the system (this is also part of the complexity of our system that we do not think would be captured by a stateless game or other gridworld setting). The drivers themselves, however, do not learn or respond to the state of the system, though we agree that this would be an interesting expansion of this work.

---

### Review · Reviewer_8qqZ · 2023-06-18

**Summary Of Contributions:**

This work proposes an approach to solve a multi-principle assistance game problem, via an explicit preference aggregation rule used to parameterize a multi-objective optimization problem which is solved by deep Q learning.  The method is tested in a multi-objective traffic setting, with the aim of studying the underlying problems of multi-principle assistance games in less artificial settings.

**Audience:**

Yes

**Claims And Evidence:**

Yes

**Requested Changes:**

1) Make it clear in the introduction how this builds in the direction of prior work on multi-principle assistance games, making it clear that the contribution is the Deep RL method and grounding to a concrete use case
2) Strengthen the literature review to foundational works of the cited fields which existed before 2016
3) Make the assumptions underlying the method clear in the text, and how these differ from or rely on assumptions of prior work, specifically:
- the assumption of utilitarian preference aggregation
- the assumption of non-strategic (truthful) agents giving preferences
- the assumption of a centralized controller and socially trusted preference aggregation rule


**Strengths And Weaknesses:**

# Strengths:
This paper is broadly well written and clear.  I understand what was done and what the results are, and it is tackling an important and understudied problem.

# Weaknesses:
There are a few key weakness I see with this work.  I have organized the below to correspond to the requested changes.

## Framing
I think it is poorly framed in that a typical reader may assume this is the first work to try to bridge social choice theory and value alignment, but there are at least two prior works.
* Multi-principal assistance games, Arnaud Fickinger, Simon Zhuang, Dylan Hadfield-Menell, Stuart Russell
* Servant of Many Masters: Shifting priorities in Pareto-optimal sequential decision-making , Critch & Russell

The first of these is cited, but is critically left out of the framing in the introduction.

In this context this paper should be re-framed to be more explicitly about the contributions with relative to these papers: he use of Deep RL and specific architectural innovations necessary to solve these problems in practice.

In addition, framing this as a solution to the reward hacking problem misses the point of that problem.  The method arrives at a reward function through other means, and though that reward may not be hackable in the original way, it will often be hackable in some new way that was not considered.

## Related Work
The related work section is very sparse and biased towards very recent ideas.  There are barely any citations before 2016, when every field involved here (IRL, multi-objective RL, social choice) are are very old fields that had been around a decade, or several decades, before that.  As such it seems that the authors are failing to cite most of the work that they are building on, and most of the important related ideas in the space.  It would be very difficult for a reader who did not already know the IRL/MORL/social choice literatures to orient this paper with respect to them.

## Unclear Assumptions
Despite taking motivation from social choice, the paper violates many of the underlying assumptions of that theory.  Most importantly individual rationality (assuming the agents are truthfully revealing their preferences).  The aggregations rules used in this work are vulnerable to an agent who lies about their preferences, and it should be stated explicitly that this is a sharp deviation from most work in social choice.

On other points, the assumptions follow the social choice literature, but as this paper takes explicit motivation from (Birhane et al., 2022), it would be best to make explicit the assumptions underlying their work as well as the remaining gap to between this work and real-world considerations.  Specifically, this work continues to uphold assumptions that there should be a centralized controller, that we should trust a centralized preference aggregation rule, and that such a rule is well evaluated on utilitarian grounds.  It is also still removed from the social context as it does not handle the practicalities such as accurate pooling of user preferences, accurately informing users.  These are not necessarily limitations of the paper, but if the goal is to correct the issues outlined in (Birhane et al., 2022) then acknowledging these gaps explicitly is an important step.

# Minor Concerns
- I don't understand the function of the word "simultaneously" in the first sentence.
- Current reward functions are "set arbitrarily", people spend a lot of time trying to get them right!
- "dynamic value alignment" is never defined, and I did not understand what it meant in the abstract.
- In the last paragraph of page 2 they mention "preference change" as a motivation for their approach.  The idea of "preference change" is a very thorny issue, and probably complicates the paper substantially in the minds of many readers.  That complexity seems avoidable in this case.
- What does it mean for a reward to be "applied to everyone".  This seems to refer to some subset of all state-based rewards but it is never made clear.
- The idea here is more closely related to the veil of ignorance than to Kant's imperative, since the agents are giving preferences not actions.
- Why are downstream agents not allowed to vote? They are all effected by the emissions!
- Rather than saying "able to avoid missalignment" in the conclusion, it would be more accurate to say "able to avoid missalignment in our setting"
- using a softmax to normalize scores is a bit weird and non-standard, as it will change the agent's preferences over some mixed lotteries.  It would be more standard to do the normalization via an affine transformation (subtract the min possible reward and divide by the max possible reward)

---

> ### Author Response · Authors · 2023-06-22
> **Response**
>
> We thank the reviewer for their review, their engagement with our work has been much appreciated. We agree with the vast majority of the comments and have implemented the changes requested by the reviewer (all changes highlighted in blue in the new version of the manuscript).
> 1. We have expanded the introduction to clarify that we build on existing work and specify the contributions as the Deep RL method and use case grounding.
> 2. We have expanded the background section to include works from before 2016 where appropriate.
> 3. We have added a subsection "Underlying assumptions" to the Methodology where we explicitly state our assumptions.
>
> One point that we would like to take up with the reviewer is that we believe that the method we propose is not prone to manipulation through misrepresentation of preferences. That is because we are not using rank-based voting schemes. This might have not been clear in the previous version. We have tried to clarify that in this version but ask the reviewer to let us know if we have succeeded. We would be happy to clarify it further as this is an important point.
>
>
> Minor concerns:
> * we have implemented changes to almost all issues listed in minor concerns except:
> 1. downstream agents not able to vote - in our simple system if the downstream agents were able to vote that would mean that the entire system is always able to vote, we wanted to make the voting population fluctuate dynamically, which we believe to be more realistic. Furthermore, in the natural extension of this system to larger networks, downstream vehicles will be upstream of a different intersection, and thus these agents will always be involved in another vote at their corresponding intersections.
> 2. we use softmax normalisation as opposed to affine transformation because the maximum and minimum possible rewards are dependent on the number of users in the system, while in our case this is known, in real systems that would not be known.
>
> Again, we thank you for your engagement and would be happy to follow up on any issues that might remain.

---

> > ### Comment · Reviewer_8qqZ · 2023-06-28
> > **Response to Response**
> >
> > Thank you for addressing many of my concerns.
> >
> > >One point that we would like to take up with the reviewer is that we believe that the method we propose is not prone to manipulation through misrepresentation of preferences. That is because we are not using rank-based voting schemes. This might have not been clear in the previous version. We have tried to clarify that in this version but ask the reviewer to let us know if we have succeeded. We would be happy to clarify it further as this is an important point.
> >
> > I agree that the general arrow's theorem doesn't apply out of the box, but generally these impossibility results apply to any way of polling agents to aggregate preferences. Usually this can be shown as a small modification of arrow's theorem, but more broadly can be inferred from the revelation principle results in mechanism design more broadly.
> >
> > >It is, however, avoided in this work by relying on binary input.
> >
> > More specifically, this claim is wrong. For instance, if you think of a "vote" as a claim that "X candidate is better than the others", then this is equivalent to using a preference aggregation rule that only looks at the top-ranked candidate for each agent and ignores all other information.  Thus arrows theorem applies to that rule and there are some situations in which the voting for the best candidate is not the strategically best thing for the agents to do.  This raises a question of how you choose what each agent should vote for, since choosing the best candidate isn't necessarily the best for them strategically you would have to solve for the Nash equilibrium to model them strategically.  Even then there is ambiguity about which equilibrium.
> >
> > In your experiments, I think you can avoid this by using only two candidates, at which point arrows theorem actually no longer applies in any form, and strategic and honest behavior are the same. This would only become a problem in a 3 or more candidate setting.
> >
> >
> > Minor concern:
> > * A few of the references in the related work (specifically in the social choice section) are not hyperlinked properly

---

> > > ### Author Response · Authors · 2023-06-28
> > > **Response to response to response**
> > >
> > > Thank you very much for your response. We understand what you mean and have edited the Underlying Assumption section to highlight that the reason we avoid the consequences of the theorem is the fact that we only have two objectives to choose from. We note that if more objectives were added, for which the system allows, the problem of potentially dis-honest strategic agent would need to be taken into account.
> > >
> > > We have also added the working hyperlinks to the indicated section.
> > >
> > > Thank you for your continued engagemend with our work, it is appreciated.

---

### Decision · Action_Editors · 2023-07-24

**Recommendation:** Reject

**Comment:**

Decision made based on my full read and assessment of the paper, in addition to the reviewers' reports and discussions with the authors. Post discussion, the reviewers are split: gaay and 8qqZ, who share similar research pov (former labmates) are learning accept, vGax recommends reject, and the fourth reviewer provided a subpar favorable decision without further engagement.

My take is that the paper would benefit from not being accepted -- the idea might be there, but the current framing and the results are not convincing (see the justification). Thus, I recommend allowing a major revision at the later time.

gaay: "The authors provide a small extension of a notion of value alignment from grid worlds to a more realistic system. While this is not a hugely significant result, it is an improvement over what is currently in the literature and will be interesting to some readers."

8qqZ: "I believe this paper is technically accurate and the experiments support that their proposed method worked for their chosen task. I think that they have fixed their literature review and framing just enough for it to be acceptable, so readers would be able to find the other related works.

I think it is harder to argue that this work is interesting, since it uses many heuristic and hand-designed choices where more established solutions exist (for instance, hand-designing a multi-objective optimization approach). ...."

vGax: "I recommend that this article be rejected due to the issues observed by me and other reviewers. The main one being that the framing of the work is lacking; and that the problem, as stated, is better suited for simpler analysis tools"

**Audience:**

Sequential decision making, HCI, and HRI communities might be interested in this work.

**Claims And Evidence:**

The paper presents a method for agent adaption to the participants values in the shared autonomy setting, based on the users preferences and a voting mechanism.

Based on the reviewer comments, the discussion with the authors, and my own assessment of the paper, in its current form the paper is not suitable for TMLR. However, the paper offers promising ideas that would benefit from the further refinement and rigor, and can be submitted in the future as a major revision.

The paper addresses a timely topic in alignment and system / user interaction. The paper presents the method thought a single, simplified example of an intersection problem. The major issue with its current form that it is not clear how the presented method can be applied or developed beyond the presented example, and the presented example is not very compelling.  For example, HCI and HRI fields have studied learning and adaptation of the user's changing preferences in more complicated settings (e.g. https://dl.acm.org/doi/pdf/10.1145/3568162.3576965)

The future version of the manuscript should present a method that empirically demonstrates the method on more than one problem setting, and compare the method against the relevant baselines.

**Resubmission Of Major Revision:**

The authors may consider submitting a major revision at a later time.